# Galectin-3 is Associated with Cardiovascular Events in Post-Acute Coronary Syndrome Patients with Type-2 Diabetes

**DOI:** 10.3390/jcm9041105

**Published:** 2020-04-13

**Authors:** Ana Lorenzo-Almorós, Ana Pello, Álvaro Aceña, Juan Martínez-Milla, Óscar González-Lorenzo, Nieves Tarín, Carmen Cristóbal, Luis M Blanco-Colio, José Luis Martín-Ventura, Ana Huelmos, Carlos Gutiérrez-Landaluce, Marta López-Castillo, Andrea Kallmeyer, Ester Cánovas, Joaquín Alonso, Lorenzo López Bescós, Jesús Egido, Óscar Lorenzo, Jose Tuñón

**Affiliations:** 1Department of Internal Medicine, IIS-Fundación Jiménez Díaz, 28040 Madrid, Spain; alorenzoa@fjd.es; 2Laboratory of Vascular Pathology, IIS-Fundación Jiménez Díaz, 28040 Madrid, Spain; lblanco@fjd.es (L.M.B.-C.); jlmartin@fjd.es (J.L.M.-V.); jegido@quironsalud.es (J.E.); olorenzo@fjd.es (Ó.L.); 3Department of Cardiology, IIS-Fundación Jiménez Díaz, 28040 Madrid, Spain; ampello@quironsalud.es (A.P.); aacena@fjd.es (Á.A.); juan.mmilla@quironsalud.es (J.M.-M.); ogonzalez@quironsalud.es (Ó.G.-L.); marta.lcastillo@fjd.es (M.L.-C.); andrea.kallmeyer@quironsalud.es (A.K.); ester.canovas@fjd.es (E.C.); 4Department of Medicine, School of Medicine, Universidad Autónoma, 28029 Madrid, Spain; 5Department of Cardiology, Hospital Universitario de Móstoles, 28935 Madrid, Spain; nieves.tarin@salud.madrid.org; 6Department of Cardiology, Hospital de Fuenlabrada, 28942 Madrid, Spain; carmen.cristobal@salud.madrid.org (C.C.); dr.gutierrez@gmail.com (C.G.-L.); 7Department of Medicine, Rey Juan Carlos University, Alcorcón, 28943 Madrid, Spain; joaquinjesus.alonso@salud.madrid.org (J.A.); llopez@fhalcorcon.es (L.L.B.); 8Centro de Investigación Biomédica en Red de Enfermedades Cardiovasculares (CIBERCV), 28040 Madrid, Spain; 9Department of Cardiology, Hospital Universitario Fundación Alcorcón, 28922 Madrid, Spain; ahuelmos@yahoo.es; 10Department of Cardiology, Hospital de Getafe, 28905 Madrid, Spain; 11Centro de Investigación Biomédica en Red de diabetes y enfermedades metabólicas asociadas (CIBERDEM), 28040 Madrid, Spain

**Keywords:** Galectin-3, cardiovascular event, stable coronary artery disease

## Abstract

Introduction: Type-2 diabetes mellitus (T2DM) is associated with early and severe atherosclerosis. However, few biomarkers can predict cardiovascular events in this population. Methods: We followed 964 patients with coronary artery disease (CAD), assessing plasma levels of galectin-3, monocyte chemoattractant protein-1 (MCP-1), and N-terminal fragment of brain natriuretic peptide (NT-proBNP) at baseline. The secondary outcomes were acute ischemia and heart failure or death. The primary outcome was the combination of the secondary outcomes. Results. Two hundred thirty-two patients had T2DM. Patients with T2DM showed higher MCP-1 (144 (113–195) vs. 133 (105–173) pg/mL, *p* = 0.006) and galectin-3 (8.3 (6.5–10.5) vs. 7.8 (5.9–9.8) ng/mL, *p* = 0.049) levels as compared to patients without diabetes. Median follow-up was 5.39 years (2.81–6.92). Galectin-3 levels were associated with increased risk of the primary outcome in T2DM patients (Hazard ratio (HR) 1.57 (1.07–2.30); *p* = 0.022), along with a history of cerebrovascular events. Treatment with clopidogrel was associated with lower risk. In contrast, NT-proBNP and MCP-1, but not galectin-3, were related to increased risk of the event in nondiabetic patients (HR 1.21 (1.04–1.42); *p* = 0.017 and HR 1.23 (1.05–1.44); *p* = 0.012, respectively), along with male sex and age. Galectin-3 was also the only biomarker associated with the development of acute ischemic events and heart failure or death in T2DM patients, while, in nondiabetics, MCP-1 and NT-proBNP, respectively, were related to these events. Conclusion: In CAD patients, galectin-3 plasma levels are associated with cardiovascular events in patients with T2DM, and MCP-1 and NT-proBNP in those without T2DM.

## 1. Introduction

Type-2 diabetes mellitus (T2DM) is one of the most important cardiovascular risk factors [1], and death rates in diabetic people are twice those of nondiabetics [2]. T2DM and prediabetes have been associated with the development of subclinical direct myocardial damage, microvascular inflammation, and endothelial dysfunction, which contribute to a higher cardiovascular risk in these patients [3]. Indeed, T2DM and prediabetes are both associated with an early atherosclerotic burden and more extensive coronary lesions [4]. Atherosclerotic plaques of diabetic people show distinctive features, such as increased lipid content, macrophage infiltration [5], more adhesive platelets [6], and more thrombogenicity, thus making them more prone to rupture [7]. As a result, the range of potential prognostic biomarkers may differ in this population as compared to patients without T2DM. In fact, we reported previously that Fibroblast Growth Factor (FGF-23), a phosphaturic hormone, predicts adverse cardiovascular outcome in patients with stable coronary artery disease (CAD) and T2DM, but not in patients without T2DM [8]. Moreover, few available biomarkers have demonstrated an ability to predict cardiovascular events in this population. 

Galectin-3, a β-galactoside-binding lectin, has been involved in a wide range of biological actions such as fibrosis, systemic inflammation, cell growth and differentiation, apoptosis, and angiogenesis [9]. Additionally, galectin-3 has been implicated in atherosclerosis and may promote insulin resistance, lipolysis, and glucose intolerance [9,10]. Recently, galectin-3 has been associated with biomarkers of β-cell dysfunction and thus has been proposed to play a more prominent role in insulin secretion than in insulin resistance due to a pro-fibrotic mechanism [11]. However, the potential contribution of galectin-3 in predicting risk of T2DM with established CAD has not been determined yet.

We followed 964 patients with CAD, testing a panel of biomarkers that comprised galectin-3 and monocyte chemoattractant protein-1 (MCP-1), both involved in inflammation and atherothrombosis, as well as the inactive N-terminal fragment of brain natriuretic peptide (NT-proBNP), which is related to heart failure. High sensitivity C-reactive protein (hs-CRP) was studied as a reference [12].

## 2. Experimental Section

### 2.1. Patients

The BACS & BAMI *(Biomarkers in Acute Coronary Syndrome & Biomarkers in Acute Myocardial Infarction)* studies included patients admitted to five hospitals in Madrid with either non-ST elevation acute coronary syndrome or ST-elevation myocardial infarction. Inclusion criteria have been defined previously [12]. Exclusion criteria were age over 85 years, coexistence of other significant cardiac disorders except left ventricular hypertrophy secondary to hypertension, coexistence of any illness or toxic habits that could limit patient survival, impossibility to perform revascularization when indicated, and subjects to whom follow-up was not possible. In order to avoid variability of findings due to an excessive heterogeneity in the intervals between the acute event and blood extraction, the investigators agreed to exclude patients that were not clinically stable the sixth day after the index event. Once patients were stable, a second plasma sample was obtained between six and twelve months after hospital admission. The present paper is a substudy of BACS & BAMI studies, and reports data from the clinical and analytic findings obtained at the time of this second plasma extraction, relating them to subsequent follow-up. 

Between July 2006 and June 2014, 2740 patients were discharged from the study hospitals with a diagnosis of non-ST elevation acute coronary syndrome or ST-elevation myocardial infarction. Of them, 1483 patients were excluded due to the following age over 85 years (16.4%), presence of disorders or toxic habits limiting survival (29.8%), impossibility to perform cardiac revascularization (9.6%), coexistence of other significant cardiopathy (5.7%), impossibility to perform follow-up (11.9%), concomitant mental disorders (4.4%), clinical instability beyond the sixth day after the index event (10.9%), refusal to participate in the study (1.5%), and impossibility of the investigators to include them (9.8%).

Of the 1257 patients included in the acute phase, 289 did not have the plasma samples 6 to 12 months later, and 4 were excluded because they developed a cancer. Thus, 964 patients had adequate plasma samples withdrawn 6 to 12 months after being discharged and were included in this study. This plasma extraction and baseline visits took place between January 2007 and December 2014. The last follow-up visits were carried out on June 2016.

### 2.2. Ethics Statement 

The research protocol conforms to the ethical guidelines of the 1975 Declaration of Helsinki as reflected in a priori approval by the human research committees of the institutions participating in this study: Fundación Jiménez Díaz, Hospital Fundación Alcorcón, Hospital de Fuenlabrada, Hospital Universitario Puerta de Hierro-Majadahonda, and Hospital Universitario de Móstoles (code 25/2007, 24th of April 2007). All patients signed informed consent documents.

### 2.3. Study Design

At baseline, clinical variables were recorded, and 12-hour fasting venous blood samples were withdrawn and collected in Ethylenediamine tetraacetic acid (EDTA). Blood samples were centrifuged at 2500 g for 10 min and plasma was stored at −80 °C. Patients were seen every year at their hospital. At the end of follow-up, the medical records were reviewed, and patient status was confirmed by telephone contact. The primary outcome was the combination of acute ischemic events (non-ST elevation acute coronary syndrome, ST-elevation myocardial infarction, stroke and transient ischemic attack) plus heart failure and all-cause mortality. The secondary outcomes were ischemic events and the composite of heart failure and death. Non-ST elevation acute coronary syndrome was defined as rest angina lasting more than 20 min in the previous 24 h, or new-onset class III-IV angina, along with transient ST depression or T wave inversion in the electrocardiogram considered diagnostic by the attending cardiologist and/or troponin elevation. ST-elevation myocardial infarction was defined as symptoms compatible with angina lasting more than 20 min and ST elevation in two adjacent leads in the electrocardiogram without response to nitroglycerin, and troponin elevation. Past myocardial infarction was diagnosed in the presence of new pathological Q waves in the electrocardiogram, along with a concordant new myocardial scar identified either by echocardiography or nuclear magnetic resonance imaging. Stroke was defined as rapid onset of a neurologic deficit attributable to a focal vascular cause lasting more than 24 h or supported by new cerebral ischemic lesions at imaging studies. A transient ischemic attack was defined as a transient stroke with signs and symptoms resolving before 24 h without cerebral acute ischemic lesions at imaging techniques. Heart failure (HF) was a clinical diagnosis made in accordance to practice guidelines. Events were adjudicated by at least two investigators of the study, along with a neurologist for cerebrovascular events. Although all events were recorded for each case, patients were excluded from the Cox regression analysis after the first event. Then, although the total number of events is also described, patients that had more than one event were computed only once for these analyses.

### 2.4. Biomarker and Analytical Studies

Plasma determinations were performed at the laboratory of Nephrology at the Gómez-Ulla hospital and at the Biochemistry Laboratory at Fundación Jiménez Díaz. The investigators who performed the laboratory studies were unaware of clinical data. Plasma concentrations of galectin-3 and MCP-1 were determined using commercially available enzyme-linked immunosorbent assay kits (DCP00, R&D systems, Minneapolis, MN, USA and BMS279/2, Bender MedSystems, Burlingame, CA, USA, respectively) following the manufacturers’ instructions. Intra- and interassay coefficients of variation were 4.6% and 5.9% for MCP-1 and 6.2% and 8.3% for galectin-3. hsCRP protein was assessed by latex-enhanced immunoturbidimetry (ADVIA 2400 Chemistry System, Siemens, Germany) and NT-proBNP by immunoassay (VITROS, Ortho Clinical Diagnostics Raritan, city, New York, NJ, USA). Lipids, glucose, and creatinine determinations were performed by standard methods (ADVIA 2400 Chemistry System, Siemens, Germany). The estimated glomerular filtration rate (eGFR) was calculated using the CKD-EPI equation. 

### 2.5. Statistics

Quantitative data with a normal distribution are presented as mean ± standard deviation (SD) and were compared using the Student *t* test. Data that did not follow a normal distribution are displayed as median (interquartile range) and compared with the Mann–Whitney test. Qualitative variables are shown as percentages and compared with the Chi-square or the Fisher’s test when appropriate. A supplemental Akaike-Information Criterion (AIC) statistic was performed for the primary event, adding to the Model 1 the effect of the different biomarkers (NT-proBNP, MCP-1, and Gal-3) (Appendix A).

The Cox proportional hazard model was used with forward stepwise selection to assess the variables associated with the outcomes. Variables with a *p* < 0.05 were considered statistically significant. 

In Model 1, risk was estimated including age, gender, diabetes, smoking status, hypertension, body mass index, history of cerebrovascular disease, atrial fibrillation, ejection fraction <40%, glomerular filtration rate assessed by Chronic Kidney Disease Epidemiology Collaboration method, lipid levels (high density lipoprotein-cholesterol (HDL-C), low density lipoprotein-cholesterol (LDL-C), triglycerides), hs-CRP, therapy with aspirin, anticoagulants, statins, angiotensin receptor blockers, angiotensin-converting enzyme inhibitors, beta-blockers, nitrates and/or nitroglycerin, diuretics, and complete revascularization. In Model 2, we included factors of the Model 1 and biomarkers NT-proBNP, MCP-1, and galectin-3 and high sensitivity Troponin-I (hs-Tr-I).

Galectin-3 and MCP-1 cutoff values were identified using a receiver operating characteristic curve (ROC) and the Youden’s index and were 6.49 ng/mL and 123.3 pg/mL, respectively. The Kaplan–Meier curve and log-rank test were used to compare time to outcome according to a multimarker score. Analysis was performed with SPSS 19.0 (IBM, Armonk, NY, USA).

## 3. Results

### 3.1. Baseline Characteristics

From 964 patients with acute coronary syndrome (ACS), 232 had T2DM and 732 were nondiabetic, with a mean age of 61.0 (54–72)- and 60.0 (51–71)-years-old, respectively, *p* = 0.092 (Table 1). There was a male predominance (75.0% vs. 76.6%, *p* = 0.609) in both groups. Compared to the nondiabetic patients, those with T2DM showed higher body mass index and had previously been diagnosed with hypertension and HF in a higher proportion.

T2DM patients had higher values of plasma triglycerides and glucose and lower total cholesterol, LDL-C, and HDL-C. They also exhibited significant higher levels of hs-CRP, MCP-1 and galectin-3 than patients without T2DM. There was a nonsignificant trend toward higher NT-proBNP and hs-Tn-I in diabetic patients.

T2DM patients showed higher percentage of implantation of drug-eluting stents and coronary artery bypass grafting at the previous acute ischemic event. Also, they had less complete revascularization compared to patients without diabetes. Finally, T2DM patients were more frequently under treatment with acetylsalicylic acid, angiotensin receptor blockers, diuretics, and nitrates. The median follow-up was 5.39 years (2.81–6.92).

### 3.2. Primary Outcome

In nondiabetic patients, NT-proBNP (HR 1.21 (1.04–1.42), *p* = 0.017) and MCP-1 (HR 1.23 (1.05–1.44), *p* = 0.012) were associated to an increased risk of developing the primary outcome, along with male sex (HR 2.03 (1.15–3.61), *p* = 0.015) and age (HR 1.04 (1.02-1.06), *p* < 0.001) (Table 2). Statins were related to a low risk (HR 0.44 (0.24–0.82), *p* = 0.010) of developing the primary event.

In contrast, in T2DM patients, higher levels of galectin-3 were associated with increased risk of developing the primary outcome (HR 1.57 (1.07–2.30), *p* = 0.022), along with a history of cerebrovascular events (HR 9.01 (2.10–38.7), *p* = 0.003). Treatment with Clopidogrel was associated with lower risk for the outcome (HR 0.44 (0.21–0.95), *p* = 0.037).

In the AIC results, the addition of the different biomarkers (NT-proBNP, MCP-1, and Gal-3) to Model 1 obtained decreased values compared with the baseline model (Appendix A).

A supplemental table regarding acute ischemic events, heart failure, or death in diabetic-hypertensive patients is also shown (Appendix A).

Galectin-3 cutoff was established at 6.49 ng/mL, and a multivariate Cox analysis was performed with this categorical variable. In the analysis, Gal-3 ≥ 6.49 ng/mL was associated with an increased risk of acute ischemic events, HF, or death (HR 3.26 (1.32–8.04), *p* = 0.010), along with a history of cerebrovascular events (HR 8.46 (2.06-34.8), *p* = 0.025) and treatment with insulin (HR 2.07 (1.06–4.02), *p* = 0.032). In contrast, statins were associated with lower risk of developing the outcome (HR 0.28 (0.09–0.89), *p* = 0.031).

In the Kaplan–Meier curve, the presence of galectin-3 ≥6.49 ng/mL in T2DM patients was significantly associated with a decrease in the number of patients who remained event-free, compared to those with galectin-3 below that value (Log rank test 10.5, *p* = 0.001) (Figure 1).

### 3.3. Acute Ischemic Events

MCP-1 was the only biomarker associated with an increased risk of acute ischemic events in the nondiabetic subgroup (HR 1.23 (1.01–1.49), *p* = 0.047), along with increased LDL-C (HR 1.01 (1.00–1.02), *p* = 0.016), male sex (HR 2.45 (1.19–5.01), *p* = 0.047), and smoking habit (HR 1.97 (1.10–3.51), *p* = 0.022) (Table 3). 

In contrast, in T2DM patients, galectin-3 was the only biomarker associated with an increase of the incidence of this outcome (HR 1.83 (1.13–2.98), *p* = 0.014), along with previous history of cerebrovascular events (HR 12.4 (2.16–71.1), *p* = 0.005). Treatment with angiotensin converting enzyme inhibitors (HR 0.35 (0.14–0.87), *p* = 0.025) and male sex (HR 0.32 (0.11–0.91), *p* = 0.032) were associated with lower risk of developing this outcome. 

### 3.4. Heart Failure or Death

NT-proBNP (HR 1.29 (1.07–1.56), *p* = 0.008) and age (HR 1.11 (1.07–1.15), *p* < 0.001) were associated with an increased risk of HF and death in nondiabetic patients (Table 4). In contrast, statins were associated with a reduced risk of this outcome (HR 0.29 (0.11–0.76), *p* = 0.012). 

However, galectin-3 was the only biomarker associated with an increased risk of developing HF and death in T2DM patients (HR 2.14 (1.18–3.91), *p* = 0.013), along with age (HR 1.10 (1.04–1.15), *p* < 0.001), atrial fibrillation (HR 8.84 (1.07–73.7), *p* = 0.043), and treatment with anticoagulants (HR 33.32 (2.46–450.5), *p* = 0.008), aldosterone receptor blockers (HR 7.59 (1.66–34.7), *p* = 0.009) and nitrates (HR 4.05 (1.20–13.6), *p* = 0.024). In contrast, LDL-c (HR 0.97 (0.95–0.99), *p* = 0.042) plasma levels were associated with lower risk. 

## 4. Discussion

In this study, galectin-3 was consistently associated with recurrent cardiovascular events (i.e., acute ischemic events, HF, or death) in a population of T2DM patients with stable CAD. In the nondiabetic subgroup, MCP-1 was associated with higher risk of acute ischemia, and NT-proBNP was related to an increased risk of HF or death.

T2DM influences the development of cardiovascular disease by contributing to CAD through atherosclerosis, and also by directly affecting the myocardium [13]. Atherosclerosis in T2DM patients is also associated with an increased inflammatory component that makes plaques more prone to complications and rupture [7]. In addition, myocardial tissue of T2DM patients shows increased fibrosis and apoptosis [3].

A body of evidence suggests a potential role of galectin-3 in HF. Galectin-3 contributes to the fibroinflammatory response by turning fibroblasts into myofibroblasts, which produce matrix proteins including collagens, fibronectin, and TGF-β [14], thus indicating that it may play a role in cardiac fibrosis and remodeling [15]. Moreover, after myocardial infarction, galectin-3 may contribute to the healing response by reducing injured cells [16]. At latter stages, however, chronic elevation of galectin-3 levels may promote tissue fibrosis, contributing to progressive adverse cardiac remodeling [15,16].

Although the relationship between galectin-3 and atherosclerosis is not completely understood, galectin-3 plays a role in the formation and destabilization of the atherosclerotic plaque [17,18]. Additionally, it may act as a chemo-attractant for monocytes to the vascular wall [19] and may also induce differentiation of macrophages into foam cells [20]. Interestingly, the highest galectin-3 levels have been found in unstable regions of atherosclerotic plaques [21,22]. Moreover, galectin-3 may promote atherosclerosis through the activation of vascular smooth muscle cells by oxidized LDL particles [23] and by inducing insulin resistance [24].

Galectin-3 was reported to be a predictor of cardiovascular events and mortality [25] in patients with HF, and it also predicts cardiovascular events in patients with CAD at high risk [26,27]. Accordingly, in the COACH study, galectin-3 levels were independent predictors of death and HF in patients with previous HF [28]. 

Galectin-3 levels are higher in T2DM patients, with or without cardiovascular disease [29,30], and our results also show increased galectin-3 values in these patients. Nevertheless, conflicting results have been found regarding the effect of galectin-3 on T2DM. For instance, higher galectin-3 levels have been related to an increased risk of diabetic retinopathy, but other research has reached the opposite conclusion [30,31]. However, galectin-3 could play a dual role in T2DM. On the one hand, it could bind and break down advanced glycation end products [32], and, on the other, it could promote insulin resistance and β-cell dysfunction [9]. Ozturk et al [33] showed that higher galectin-3 levels predicted coronary atherosclerosis in 158 T2DM patients studied using coronary computed tomography angiography. Also, increased galectin-3 levels have been associated with all-cause mortality in T2DM patients with low [34] and high [35] cardiovascular risk, even without prior CVD and independently of traditional risk factors. 

Seferovic et al. demonstrated that elevated galectin-3 levels correlated with NT-proBNP and left ventricular mass in patients with T2DM and hypertension without ischemic or HF symptoms, suggesting the potential role of galectin-3 in the detection of early myocardial structural remodeling [36]. In our study, 81.9% of T2DM patients were also hypertensive, compared to only 58.6% in the nondiabetic group. However, hypertension was not predictive of outcomes in our results. The potential presence of myocardial structural remodeling in asymptomatic patients and, thus, early subclinical cardiac disease in T2DM patients, could partially explain the higher levels of galectin-3 found in the diabetic group. 

Of interest, aldosterone-receptor blockers were associated with increased risk of HF or death in the T2DM subgroup, as were anticoagulants and nitrates. This finding likely reflects the fact that patients in whom these drugs are indicated have worse prognosis than those who do not require these therapies. 

In the *Aggrastat to Zocor trial* [37], higher levels of MCP-1 were related to increased risk of death, myocardial infarction, or cardiovascular events on follow-up. These results were consistent with previous data from our group [12]. In our study, NT-proBNP was associated with HF or death in the nondiabetic patients, though the same was not true for the T2DM population. This could be due in part to the limited sample size of the T2DM sub-group. Despite this limitation, galectin-3 was a significantly associated with of outcomes in this population.

There are scarce publications suggesting the potential role of MCP-1, galectin-3, and NT-proBNP to predict future cardiovascular events in stable CAD patients [38]. More studies are needed to clarify the selective role of galectin-3 in T2DM and to predict recurrent events in CAD. However, according to these results, galectin-3 shows promise as a biomarker in T2DM. 

### Limitations

The exclusion of patients who were unstable within the first days after the ACS event may have biased our results, as these patients probably have a worse prognosis. However, only 10.9% of cases were excluded for this reason. Second, the limited size of the diabetic population included in this study may have influenced the results. There is no consensus regarding galectin-3 cutoff and, therefore, the use of different values may have led to different results. Finally, potential fluctuations in galectin-3 and MCP-1 levels during follow-up have not been explored.

## 5. Conclusions

Galectin-3 is associated with increased risk of cardiovascular events in diabetic patients with stable CAD. In patients without T2DM, in contrast, MCP-1, and NT-proBNP are associated with the risk of acute ischemia and HF or death, respectively.

## Figures and Tables

**Figure 1 jcm-09-01105-f001:**
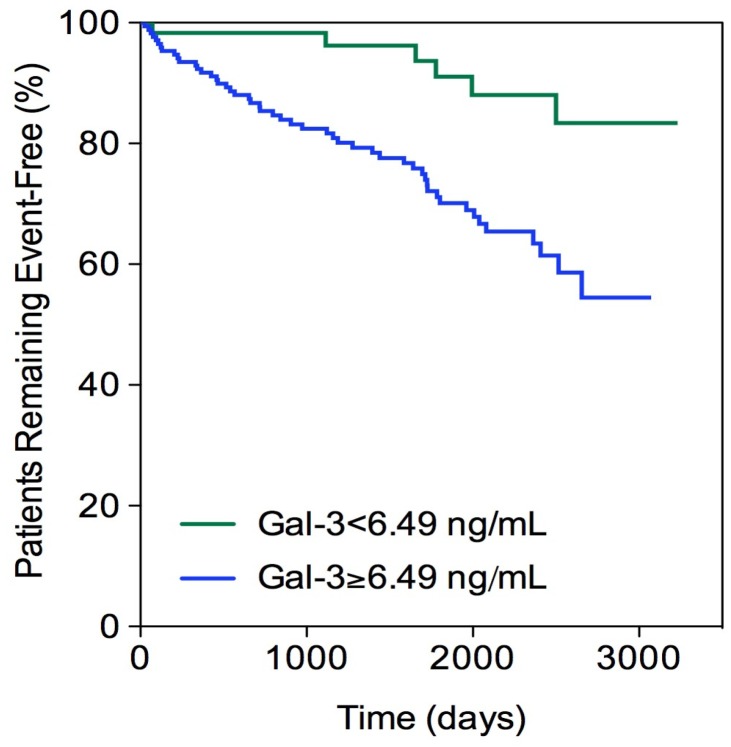
Kaplan–Meier curve. Diabetic patients. Total events = 56. Galectin-3 (*Gal-3*).

**Table 1 jcm-09-01105-t001:** Characteristics of patients with and without diabetes.

Characteristic	Patients without Diabetes (N = 732)	Patients with Diabetes (N = 232)	*p*-Value
Age (yr)	60.0 (51–71)	61 (54–72)	0.092
Male sex (%)	76.6	75.0	0.609
Body mass index (Kg/m^2^)	27.7 (25.3–30.3)	29.0 (26.9–32.1)	**<0.001**
Present Smoker (%)	14.1	13.4	0.786
Hypertension (%)	58.6	81.9	**<0.001**
Previous heart failure (%)	9.0	19.8	**<0.001**
Cerebrovascular events (%)	2.3	4.3	0.110
Present or past atrial fibrillation (%)	6.7	5.2	0.407
Total cholesterol (mg(dL)	145 (126–165)	140 (121–157)	**0.010**
LDL cholesterol (mg/dL)	79 (65–93)	73 (62–88)	**0.005**
HDL cholesterol (mg/dL)	42 (35–48)	38 (33–45)	**<0.001**
Triglycerides (mg/dL)	97 (75–135)	113 (82–161)	**<0.001**
Glucose (mg/dL)	97 (90–106)	130 (109–159)	**<0.001**
GFR (CKD-EPI) (mL/min/1.73 m^2^)	81 (67–93)	77 (63–91)	0.078
High-sensitivity C-reactive protein (mg/L)	1.1 (0.3–2.9)	1.5 (0.6–4.5)	**0.002**
NT-ProBNP (pg/mL)	168 (89–381)	193 (101–473)	0.103
MCP-1 (pg/mL)	133 (105–173)	144 (113–195)	**0.006**
Galectin-3 (ng/mL)	7.8 (5.9–9.8)	8.3 (6.5–10.5)	**0.049**
High-sensitivity troponin I (ng/mL)	0.003 (0.000–0.010)	0.004 (0.000-0.011)	0.174
Medical Therapy			
Acetylsalicylic acid (%)	92.5	97.0	**0.015**
AntiP2Y12 (%)	74.9	77.2	0.480
Acenocumarol (%)	5.3	6.0	0.681
Statins (%)	95.1	94.0	0.504
Oral antidiabetic drugs (%)	0	71.1	**<0.001**
Insulin (%)	0	27.6	**<0.001**
ACE inhibitors (%)	62.3	63.4	0.770
Angiotensin receptor blockers (%)	13.9	19.8	**0.030**
Aldosterone receptor blockers (%)	6.1	8.6	0.191
Betablockers (%)	79.1	78.9	0.943
Diuretics (%)	15.4	29.3	**<0.001**
Nitrates (%)	11.2	18.5	**0.004**
Data at Last Acute Coronary Event			
STEMI/Non-STEACS (%)	48.7/51.3	55.6/44.4	0.070
Number of vessels diseased	1.0 (1.0, 2.0)	1.0 (1.0, 2.0)	**<0.001**
Drug-eluting stent (%)	49.0	62.9	**<0.001**
Coronary artery bypass graft (%)	4.5	9.0	**0.009**
Complete revascularization (%)	73.2	60.8	**<0.001**

Results expressed as media ± SD or median (IQR). Significant results expressed in bold characters. BMI: Body mass index; LDL: Low-density lipoprotein; HDL: High-density lipoprotein; CKD-EPI: Chronic Kidney Disease Epidemiology Collaboration; GFR: Glomerular filtration rate; MCP-1: Monocyte chemoattractant protein-1; Non-STEACS: Non-ST elevation acute coronary syndrome; NT-Pro-BNP: Pro-Brain natriuretic peptide; STEMI: ST-elevation myocardial infarction; ACE: Angiotensin-converting enzyme.

**Table 2 jcm-09-01105-t002:** Cox proportional hazards model for the incidence of acute ischemic events, heart failure or death.

	No Diabetic Patients		Diabetic Patients	
**Variable**	**Model 1**	Model 2	Model 1	Model 2
	HR (95% CI)	HR (95% CI)	HR (95% CI)	HR (95% CI)
Age, years	**1.04 (1.02–1.07)**	**1.04 (1.02–1.06)**	1.02 (0.99–1.06)	1.03 (0.99–1.06)
Sex, male	**1.83 (1.05–3.16)**	**2.03 (1.15–3.61)**	**0.38 (0.17–0.86)**	0.44 (0.18-1.04)
Smoker, yes	1.33 (0.70–2.56)	0.78 (0.47–1.28)	1.82 (0.73–4.57)	2.29 (0.84–6.25)
Hypertension, yes	1.42 (0.88–2.30)	1.36 (0.83–3.69)	1.46 (0.53–4.00)	1.79 (0.64–4.95)
Body mass index, kg/m^2^	1.00 (1.00–1.00)	1.00 (1.00–1.00)	0.99 (0.97–1.01)	0.99 (0.97–1.01)
History of CVE, yes	0.84 (0.24–2.91)	1.06 (0.31–3.71)	**9.62 (2.47–37.6)**	**9.01 (2.10–38.7)**
Ejection fraction <40%, yes	**2.11 (1.21–3.69)**	1.66 (0.89–3.07)	0.62 (0.26–1.50)	0.68 (0.27–1.67)
Atrial Fibrillation, yes	0.79 (0.32–1.89)	0.70 (0.27–1.81)	3.33 (0.99–11.14)	3.27 (0.91–11.83)
Acute myocardial infarction, yes	0.84 (0.24–2.91)	0.67 (0.43–1.03)	1.19 (0.57–2.46)	1.10 (0.53–2.29)
Complete Revascularization	0.89 (0.58–1.35)	0.87 (0.57–1.34)	0.75 (0.38–1.48)	0.85 (0.43–1.67)
LDL-c, mg/dL	1.00 (0.99–1.01)	1.00 (1.00–1.01)	1.00 (0.99–1.01)	1.00 (0.99–1.01)
HDL-c, mg/dL	1.00 (0.98–1.02)	1.00 (0.98–1.02)	1.01 (0.97–1.05)	1.01 (0.97–1.05)
Triglycerides, mg/dL	1.00 (1.00–1.00)	1.00 (1.00–1.00)	1.00 (1.00–1.00)	1.00 (1.00–1.01)
CKD-EPI <60 mL/min/1.73 m^2^	0.96 (0.59–1.57)	0.75 (0.44–1.27)	0.86 (0.40–1.84)	0.77 (0.34–1.71)
Acetylsalicylic acid, yes	1.20 (0.59–2.42)	1.11 (0.54–2.25)	0.66 (0.14–3.08)	0.84 (0.18–3.93)
AntiP2Y12, yes	0.96 (0.63–1.45)	0.98 (0.64–1.51)	**0.42 (0.20–0.87)**	**0.44 (0.21–0.95)**
Anticoagulants, yes	1.19 (0.50–2.82)	1.00 (0.39–2.53)	2.65 (0.65–10.7)	2.81 (0.64–12.2)
Statins, yes	**0.45 (0.25–0.82)**	0.44 (0.24–0.82)	**0.32 (0.11–0.98)**	0.37 (0.98–3.89)
ACE inhibitors, yes	0.89 (0.56–1.43)	**0.92 (0.56–1.49**)	0.49 (0.23–10.4)	0.49 (0.22–1.08)
ARB, yes	1.12 (0.63–2.00)	1.16 (0.63–2.11)	0.95 (0.40–2.26)	0.82 (0.32–2.07)
Anti-aldosterone, yes	0.79 (0.36–1.77)	0.76 (0.33–1.75)	2.62 (0.95–7.20)	2.12 (0.75–6.04)
β-Blockers, yes	0.86 (0.55–1.36)	0.84 (0.52–1.34)	0.98 (0.48–2.01)	0.92 (0.45–1.86)
Nitrates, yes	1.53 (0.93–2.53)	1.34 (0.79–2.26)	1.98 (0.93–4.20)	1.83 (0.86–3.90)
Diuretics, yes	1.19 (0.74–1.92)	1.14 (0.70–1.88)	1.04 (0.50–2.14)	0.89 (0.40–1.95)
Insulin, yes	-	-	1.89 (1.00–3.60)	1.95 (0.98–3.89)
Oral antidiabetic drugs, yes	-	-	0.98 (0.49–1.97)	1.01 (0.48–2.11)
Hs-CRP, mg/L	0.98 (0.95–1.01)	0.98 (0.95–1.02)	0.97 (0.93–1.01)	0.96 (0.92–1.01)
NT-proBNP, 1-SD	-	**1.21 (1.04–1.42)**	-	1.08 (0.81–1.44)
MCP-1, 1-SD	-	**1.23 (1.05–1.44)**	-	0.93 (0.61–1.41)
Gal-3, 1-SD	-	1.07 (0.89–1.28)	-	**1.57 (1.07–2.30)**
Hs-cTnT, 1-SD	-	1.42 (0.49–4.09)	-	0.95 (0.81–1.12)

Model 1 was adjusted by age; gender; smoking status; hypertension; body mass index; low-density lipoprotein (LDL-C), high-density lipoprotein (HDL-C), and triglyceride plasma levels; history of cerebrovascular events (CVE), ejection fraction <40%, or atrial fibrillation; glomerular filtration rate assessed by Chronic Kidney Disease Epidemiology Collaboration method <60 (CKD-EPI); high-sensitivity C-reactive protein (Hs-CRP); therapy with aspirin, clopidogrel, anticoagulants, antiP2Y12, statins, angiotensin-converting enzyme (ACE) inhibitors, angiotensin receptor blockers (ARB), anti-aldosterone, β-blockers, nitrates, and/or nitroglycerin; diuretic use; and type of last acute coronary event or existence of complete revascularization at the event. In diabetic patients, Model 1 also included therapy with insulin or oral antidiabetic drugs. Model 2 risk was adjusted for factors in Model 1 and N-terminal-Probrain natriuretic peptide (NT-proBNP), Monocyte chemoattractan protein-1 (MCP-1), galectin-3 (Gal-3), and High sensitivity-TroponinT (hs-Tn). Standard deviation (SD). Significant results are expressed in bold characters.

**Table 3 jcm-09-01105-t003:** Cox proportional hazards model for the incidence of acute ischemic events.

	Nondiabetic Patients		Diabetic Patients	
**Variable**	**Model 1**	Model 2	Model 1	Model 2
	HR (95% CI)	HR (95% CI)	HR (95% CI)	HR (95% CI)
Age, years	1.01 (0.99–1.04)	1.01 (0.98–1.04)	0.99 (0.94–1.03)	0.98 (0.94–1.03)
Sex, male	**2.28 (1.14–4.56)**	**2.45 (1.19.5.01)**	**0.30 (0.11–0.82)**	**0.32 (0.11–0.91)**
Smoker, yes	**1.90 (1.08–3.35)**	**1.97 (1.10–3.51)**	1.30 (0.41–4.19)	1.71 (0.47–6.15)
Hypertension, yes	1.46 (0.81–2.63)	1.42 (0.78–2.56)	3.08 (0.77–12.3)	3.45 (0.87–13.7)
Body mass index, kg/m^2^	1.00 (0.94–1.06)	0.99 (0.93–1.06)	0.99 (0.96–1.02)	0.99 (0.95–1.02)
History of CVE, yes	0.00 (0.00–1.5 × 10^26^)	0.00 (0.00–1.8 × 10^19^)	**13.4 (2.60–69.4)**	**12.4 (2.16–71.1)**
Ejection fraction <40%, yes	2.03 (0.95–4.36)	1.75 (0.79–3.87)	0.56 (0.18–1.77)	0.63 (0.19–2.03)
Atrial Fibrillation, yes	0.46 (0.11–1.96)	0.48 (0.12–2.03)	2.01 (0.39–10.3)	2.11 (0.34–13.0)
Acute myocardial infarction, yes	0.60 (0.35–1.03)	0.62 (0.36–1.06)	0.61 (0.24–1.55)	0.57 (0.22–1.47)
Complete Revascularization	0.73 (0.44–1.21)	0.70 (0.42–1.16)	0.61 (0.26–1.45)	0.69 (0.28–1.70)
LDL-C, mg/dL	**1.01 (1.00–1.02)**	**1.01 (1.00–1.02)**	1.00 (0.99–1.02)	1.00 (0.99–1.02)
HDL-C, mg/dL	0.99 (0.96–1.02)	0.99 (0.96–1.02)	1.01 (0.97–1.06)	1.01 (0.97–1.06)
Triglycerides, mg/dL	1.00 (1.00.1.01)	1.00 (1.00–1.01)	**1.00 (1.00–1.01)**	**1.00 (1.00–1.01)**
CKD-EPI <60 mL/min/1.73 m^2^	1.07 (0.55–2.08)	0.93 (0.47–1.84)	0.86 (0.31–2.37)	0.93 (0.32–2.71)
Acetylsalicylic acid, yes	1.48 (0.53–4.11)	1.43 (0.52–3.93)	1.29 (0.12–13.4)	1.57 (0.15–16.5)
AntiP2Y12, yes	1.06 (0.62–1.81)	1.14 (0.66–1.96)	0.49 (0.20–1.23)	0.60 (0.22–1.62)
Anticoagulants, yes	1.43 (0.39–5.26)	1.31 (0.34–5.08)	1.01 (0.20–5.22)	0.87 (0.16–4.84)
Statins, yes	0.67 (0.30–1.50)	0.68 (0.30–1.54)	0.49 (0.14–1.73)	0.61 (0.15–2.53)
ACE inhibitors, yes	0.80 (0.45–1.41)	0.75 (0.42–1.33)	**0.34 (0.14–0.83)**	**0.35 (0.14–0.87)**
ARB, yes	1.38 (0.71–2.71)	1.26 (0.63–2.51)	0.77 (0.26–2.26)	0.57 (0.17–1.82)
Anti-aldosterone, yes	1.06 (0.38–2.93)	0.88 (0.30–2.58)	1.31 (0.33–5.23)	1.03 (0.24–4.40)
β-Blockers, yes	1.10 (0.59–2.06)	1.16 (0.61–2.20)	0.99 (0.42–2.33)	0.87 (0.36–2.06)
Nitrates, yes	1.42 (0.75–2.67)	1.27 (0.66–2.44)	1.33 (0.53–3.34)	1.10 (0.42–2.87)
Diuretics, yes	0.93 (0.49–1.75)	0.99 (0.52–1.88)	0.77 (0.31–1.94)	0.66 (0.23–1.89)
Insulin, yes	-	-	**2.37 (1.05–5.34)**	2.29 (0.95–5.53)
Oral antidiabetic drugs, yes	-	-	0.70 (0.30–1.60)	0.73 (0.30–1.80)
Hs-CRP, mg/L	0.97 (0.92–1.03)	0.96 (0.91–1.02)	0.97 (0.93–1.02)	0.97 (0.93–1.03)
NT-proBNP, 1-SD	-	1.15 (0.89–1.47)	-	0.95 (0.68–1.33)
MCP-1, 1-SD	-	**1.23 (1.01–1.49)**	-	0.78 (0.44–1.38)
Gal-3, 1-SD	-	1.07 (0.87–1.31)	-	**1.83 (1.13–2.98)**
Hs-cTnT, 1-SD	-	1.18 (0.35–3.96)	-	1.02 (0.84–1.23)

Model 1 was adjusted by age; gender; smoking status; hypertension; body mass index; low-density lipoprotein (LDL-C), high-density lipoprotein (HDL-C), and triglyceride plasma levels; history of cerebrovascular events (CVE), ejection fraction <40%, or atrial fibrillation; glomerular filtration rate assessed by Chronic Kidney Disease Epidemiology Collaboration method <60 (CKD-EPI); high-sensitivity C-reactive protein (Hs-CRP); therapy with aspirin, clopidogrel, antiP2Y12, anticoagulants, statins, angiotensin-converting enzyme (ACE) inhibitors, angiotensin receptor blockers (ARB), anti-aldosterone, β-blockers, nitrates, and/or nitroglycerin; diuretic use; and type of last acute coronary event or existence of complete revascularization at the event. In diabetic patients, Model 1 also included therapy with insulin or oral antidiabetic drugs. Model 2 risk was adjusted for factors in Model 1 and N-terminal probrain natriuretic peptide (NT-proBNP), Monocyte chemoattractant protin-1 (MCP-1), Galectin-3 (Gal-3), and high sensitivity-TroponinT (hs-Tn). Standard deviation (SD). Significant results are expressed in bold characters.

**Table 4 jcm-09-01105-t004:** Cox proportional hazards model for the incidence of heart failure or death.

	Nondiabetic Patients		Diabetic Patients	
**Variable**	**Model 1**	Model 2	Model 1	Model 2
	HR (95% CI)	HR (95% CI)	HR (95% CI)	HR (95% CI)
Age, years	**1.12 (1.08–1.15)**	**1.11 (1.07–1.15)**	**1.10 (1.05–1.16)**	**1.10 (1.04–1.15)**
Sex, male	1.27 (0.53–3.05)	1.50 (0.60–3.78)	0.57 (0.16–1.97)	0.87 (0.22–3.40)
Smoker, yes	0.49 (0.22–1.10)	0.57 (0.24–1.33)	2.58 (0.70–9.55)	3.55 (0.79–15.9)
Hypertension, yes	1.31 (0.60–2.85)	1.19 (0.53–2.68)	1.50 (0.28–8.08)	2.02 (0.38–10.8)
Body mass index, kg/m^2^	1.00 (1.00–1.00)	1.00 (1.00–1.00)	1.01 (0.98–1.03)	1.00 (0.97–1.03)
History of CVE, yes	2.30 (0.57–9.18)	2.70 (0.68–10.8)	**10.2 (1.31–80.2)**	10.0 (0.98–102.8)
Ejection fraction <40%, yes	**3.31 (1.54–7.11)**	2.14 (0.89–5.12)	0.34 (0.08–1.45)	0.48 (0.12–1.96)
Atrial Fibrillation, yes	1.00 (0.33–3.05)	0.70 (0.21–2.36)	**6.49 (1.04–40.4)**	**8.84 (1.07–73.7)**
Acute myocardial infarction, yes	1.00 (0.52–1.93)	0.86 (0.43–1.69)	2.38 (0.78–7.19)	2.02 (0.66–6.15)
Complete Revascularization	0.71 (0.37–1.38)	0.65 (0.33–1.30)	1.43 (0.50–4.05)	1.31 (0.47–3.66)
LDL-C, mg/dL	0.99 (0.98–1.01)	0.99 (0.98–1.01)	**0.97 (0.95–0.99)**	**0.97 (0.95–0.99)**
HDL-C, mg/dL	1.04 (0.98–1.07)	1.03 (0.98–1.05)	1.00 (0.95–1.05)	0.99 (0.94–1.04)
Triglycerides, mg/dL	1.00 (0.99–1.00)	1.00 (0.99–1.00)	1.00 (0.99–1.01)	1.00 (0.99–1.01)
CKD-EPI <60 mL/min/1.73 m^2^	0.85 (0.44–1.65)	0.99 (0.98–1.01)	1.33 (0.50–3.54)	0.86 (0.29–2.50)
Acetylsalicylic acid, yes	0.95 (0.40–2.27)	0.73 (0.29–1.82)	0.31 (0.04–2.48)	0.44 (0.05–3.83)
AntiP2Y12, yes	0.93 (0.49–1.76)	0.94 (0.49–1.82)	0.43 (0.13–1.35)	0.36 (0.10–1.22)
Anticoagulants, yes	1.36 (0.47–4.00)	1.12 (0.33–3.79)	**15.9 (1.62–157.0)**	**33.32 (2.46-450.5)**
Statins, yes	**0.35 (0.14–0.84)**	**0.29 (0.11–0.76)**	0.90 (0.14–5.55)	0.84 (0.10–7.21)
ACE inhibitors, yes	1.05 (0.48–2.27)	1.30 (0.54–3.12)	0.98 (0.29–3.25)	0.72 (0.20–2.59)
ARB, yes	0.40 (0.14–1.14)	0.51 (0.17–1.55)	1.13 (0.29–4.47)	0.74 (0.16–3.52)
Anti-aldosterone, yes	0.47 (0.15–1.44)	0.60 (0.18–1.99)	**10.2 (2.82–39.9)**	**7.59 (1.66–34.7)**
β-Blockers, yes	0.64 (0.34–1.23)	0.60 (0.30–1.18)	0.88 (0.32–2.46)	0.90 (0.32–2.57)
Nitrates, yes	1.02 (0.44–1.65)	0.83 (0.36–1.91)	**3.86 (1.16–12.8)**	**4.05 (1.20–13.6)**
Diuretics, yes	**2.02 (1.05–3.87)**	1.59 (0.78–3.24)	1.84 (0.56–6.02)	1.32 (0.37–4.72)
Insulin, yes	-	-	**2.82 (1.03–7.72)**	2.59 (0.81–8.31)
Oral antidiabetic drugs, yes	-	-	2.25 (0.71–7.14)	1.88 (0.53–6.75)
Hs-CRP, mg/L	0.99 (0.95–1.03)	1.00 (0.96–1.03)	0.97 (0.92–1.02)	0.96 (0.90–1.02)
NT-proBNP, 1-SD	-	**1.29 (1.07–1.56)**	-	1.29 (0.91–1.83)
MCP-1, 1-SD	-	1.14 (0.90–1.45)	-	0.89 (0.47–1.67)
Gal-3, 1-SD	-	1.22 (0.96–1.56)	-	**2.14 (1.17–3.91)**
Hs-cTnT, 1-SD	-	2.45 (0.73–8.21)	-	0.96 (0.78–1.17)

Model 1 was adjusted by age; gender; smoking status; hypertension; body mass index; low-density lipoprotein (LDL-C), high-density lipoprotein (HDL-C), and triglyceride plasma levels; history of cerebrovascular events (CVE), ejection fraction <40%, or atrial fibrillation; glomerular filtration rate assessed by Chronic Kidney Disease Epidemiology Collaboration method <60 (CKD-EPI); high-sensitivity C-reactive protein (Hs-CRP); therapy with aspirin, clopidogrel, antiP2Y12, anticoagulants, statins, angiotensin-converting enzyme (ACE) inhibitors, angiotensin receptor blockers (ARB), anti-aldosterone, β-blockers, nitrates, and/or nitroglycerin; diuretic use; and type of last acute coronary event or existence of complete revascularization at the event. In diabetic patients, Model 1 also included therapy with insulin or oral antidiabetic drugs. Model 2 risk was adjusted for factors in Model 1 and N-terminal probrain natriuretic peptide (NT-proBNP), Monocyte chemoattractant protin-1 (MCP-1), Galectin-3 (Gal-3), and high sensitivity-TroponinT (hs-Tn). Standard deviation (SD). Significant results are expressed in bold characters.

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
