# Peer review of "Galectin-3 is Associated with Cardiovascular Events in Post-Acute Coronary Syndrome Patients with Type-2 Diabetes"

_jcm, 2020, doi:10.3390/jcm9041105_

Round 1

Reviewer 1 Report

Authors have satisfied reviewer's comments, and the paper has been improved.

Further points that need to be discussed: 

1) With stats performed, it is more correct to say that galectin-3 is associated with outcome, rather than 'predict'. Please, amend title and text accordingly

2) It is not clear from the text which samples are you using for the analyses. The samples from baseline after the acute coronary syndrome, or the samples  taken between six and twelve months after hospital admission? Please clarify. If the the situation is the first, why do you add the sentence about the samples?

Author Response

  • With stats performed, it is more correct to say that galectin-3 is associated with outcome, rather than 'predict'. Please, amend title and text accordingly

Thank you very much for the appreciation. We have changed it in the title and text when appropriate.

2)     It is not clear from the text which samples are you using for the analyses. The samples from baseline after the acute coronary syndrome, or the samples taken between six and twelve months after hospital admission? Please clarify. If the situation is the first, why do you add the sentence about the samples?

Thank you for the suggestion. Patients were included in the present study at the time of second plasma extraction, 6-12 months after the index acute coronary syndrome, when they may be considered stable patients. Follow-up started at this point.

We have changed the text in order to make this easier to understand.

Reviewer 2 Report

Improved a lot. 

A minor comment to make it more succinct:

For result presentation form, we'd report 95% CI or p value, not both as they are duplicative. Suggest deleting the column of P value and star-label significant findings with p<0.05 in the column of HR (95% CI).

Author Response

1)       For result presentation form, we'd report 95% CI or p value, not both as they are duplicative. Suggest deleting the column of P value and star-label significant findings with p<0.05 in the column of HR (95% CI).

Thank you very much for the suggestion. In this new version we have deleted “p” values in Tables 2-4 and in the Suppl Table 2.

Reviewer 3 Report

Thank you for revising your manuscript. Authors have satisfied reviewer's comments.

Author Response

Thank you very much

This manuscript is a resubmission of an earlier submission. The following is a list of the peer review reports and author responses from that submission.

Round 1

Reviewer 1 Report

In their study, the authors describe the predictive potential of galectin-3 in patients with T2 diabetes regarding the occurrence of cardiovascular events such as ischemic events, heart failure or death. The paper is well written. In view of individualized medicine, rather than single biomarker models, multi-biomarker approaches should be taken into account by the authors. Results would thus be more appealing and significant if the authors would examine the benefit of combined models of the three markers NT-proBNP, MCP-1, and galectin-3. Additionally, the manuscript should be revised regarding English spelling and grammar.

Detailed comments:

Page 2 – line 64: “We have followed…” / line 76: “…in whom…” instead “…to whom…” / line 79: “…a second plasma sample was obtained between six and twelve months later, when the patients were stable.” Better: In stable patients, a second plasma sample was obtained between six and twelve months after hospital admission.” / line 81: 2740 should be 2,740 etc.

Page 4 – line 161: ACS – abbreviation should be introduced / line 164: “…had been previously diagnosed…” better: “…have previously been diagnosed…”

Page 5 – table 1: I would rather prefer the use of NSTEMI instead of Non-STEACS.

Page 5 – table 2: The authors show hazard ratios of different parameters in non-diabetic and diabetic patients. I wonder how the HRs for NT-proBNP and MCP-1 are in diabetic patients. Moreover, a z-transformation prior to Cox analysis allows for a direct comparison of the HRs of NT-proBNP, MCP-1, and galectin-3. This issue should be addressed in the manuscript. Additionally, in the table, it should be NT-proBNP, not Pro-BNP.

Page 6 – line 202: Authors describe the use of ROC analysis to determine the cut-off for galectin-3. Why did the authors not use Harrell’s c instead, since it appears to me to be more accurate in prediction of survival models? Please explain.

Page 7 – table 3/4: Authors describe different HRs for the incidence of acute ischemic events and heart failure or death, respectively. Both tables lack a caption. The mention of further predictive variables seems not so interesting to me. I would be more interested in the additive value of galectin-3 to NT-proBNP and MCP-1.  Meaning e.g. an AIC (Akaike-Information-Criterion) statistic regarding the combination of the biomarker (AIC_model1+ NT-proBNP vs. AIC_model1+ galectin-3 vs. AIC_model1+ NT-proBNP + galectin-3).

This manuscript needs major revision regarding content and minor revisions concerning English grammar and spelling.

Reviewer 2 Report

Interesting topic with interesting findings. My comments as below:

Title: GALECTIN-3 PREDICTS CARDIOVASCULAR EVENTS IN post-acute coronary syndrome PATIENTS WITH TYPE-2 DIABETES

Abstract: Line 27-29. Would delete such detailed demographics from abstract.

Manuscriot:

Line 44-45: ", with....being at risk". Would delete this inaccurate statement--everyone is at risk for cardiovascular disease.

Line 139-140: the reasons for choosing these cut-off values are missing. If there is no well-established cut-off values, sensitivity analyses should be performed to ensure the robustness of the study results.

Tables 2-4. It is an incomplete result table if it only reports significant results. P values >0.05 are informative too (e.g. marginally significant). If it exceeds the page limit, consider moving 1~2 tables into supplementary documents.

Minor comments: Table 2, column 5, row 5 seems to be a comma typo "0.015"

Line 271-272: In such case, could the authors provide Cox regression results for hypertensive diabetic patients as a supplementary table? Since the fact that hypertension was not a predictor might merely because there are too many hypertensive patients (81.9%).

Reviewer 3 Report

The present paper is a is a sub-study of  the BACS (Biomarkers in Acute Coronary Syndrome) & BAMI (Biomarkers in Acute Myocardial  Infarction) studies.

The idea to compare biomarkers in diabetics vs non diabetics patients with regard to cardiovascular outcome is of a certain interest. However, my impression reading this paper is that authors just showed the finding obtained using stats, without having a real hypothesis driving the paper, neither an explanation of the findings. For example, in the results section, authors showed that galectin-3 was the only biomarker associated with an increased risk of developing HF and death in T2DM patients, along with aldosterone receptor blockers, without taking into account that this association can be affected by the fact that one of the main indication to aldosterone receptor blockers treatment is actually HF. Furhter, authors stated that ‘In contrast, LDL-c plasma levels were associated  with lower risk (of developing HF and death). This is contrast with all the current literature on the topic.

All these issues are not explained in discussion, that needs to be more deep and precise.